# Relationship between Dietary Fiber Intake and the Prognosis of Amytrophic Lateral Sclerosis in Korea

**DOI:** 10.3390/nu12113420

**Published:** 2020-11-07

**Authors:** Haelim Yu, Seung Hyun Kim, Min-Young Noh, Sanggon Lee, Yongsoon Park

**Affiliations:** 1Department of Food and Nutrition, Hanyang University, 222 Wangsimni-ro, Seongdong-gu, Seoul 04763, Korea; yhr5669@naver.com; 2Department of Neurology, Hanyang University Hospital, 222 Wangsimni-ro, Seongdong-gu, Seoul 04763, Korea; kimsh1@hanyang.ac.kr (S.H.K.); nmy@hanyang.ac.kr (M.-Y.N.); sanggonlee.nr@gmail.com (S.L.)

**Keywords:** amyotrophic lateral sclerosis, dietary fiber, gut microbiota, prognosis, vegetable fiber

## Abstract

The gut microbiota has been suggested as an important factor in the pathogenic mechanisms of amyotrophic lateral sclerosis (ALS). This study aimed to investigate whether the intake of different kinds of dietary fiber was related to the disease progression rate (∆FS) and survival time. In total, 272 Korean sporadic ALS patients diagnosed according to the revised EI Escorial criteria were recruited starting in March 2011 and were followed until the occurrence of events or the end of September 2020. The events included percutaneous endoscopic gastrostomy, tracheostomy, and death. Dietary fiber intake was calculated based on a 24-h dietary recall and classified according to five major fiber-rich foods: vegetables, fruits, grains, legumes, and nuts/seeds. Among the total participants, the group with ∆FS values lower than the mean ∆FS (0.75) was noted in the highest tertiles of total and vegetable fiber intake. Participants in the highest tertile for vegetable fiber intake showed longer survival in the Kaplan–Meier analysis (*p* = 0.033). Notably, vegetable fiber intake was negatively correlated with pro-inflammatory cytokine (interleukin (IL)-1β, IL-6, and monocyte chemoattractant protein-1) levels in the cerebrospinal fluid. This study showed that vegetable fiber intake could influence the disease progression rate and survival time. Further clinical trials are needed to confirm whether dietary fiber supplementation improves the prognosis of ALS.

## 1. Introduction

Amyotrophic lateral sclerosis (ALS) is a neurodegenerative disease characterized by the progressive degeneration of motor neurons, leading to dysphagia, limb paralysis, and death due to respiratory failure [1]. The majority of ALS cases are sporadic, accounting for 90–95% of all cases, and the remaining 5–10% of cases involve familiar ALS. Despite an advanced understanding of the pathogenic mechanisms of ALS, no reliable treatment has been identified [2].

Recently, the gut microbiota has been suggested to play a role in the pathophysiology of ALS by fermenting dietary fiber and producing short-chain fatty acids (SCFAs) [3]. Previous studies have reported that patients with ALS had intestinal dysbiosis and reduced levels of butyrate-producing bacteria compared with healthy controls [4,5]. In ALS mice treated with butyrate, the main SCFA, the gut microbiome was observed to be restored and disease progression was delayed, suggesting that butyrate could modulate intestinal homeostasis in ALS [6]. Additionally, Paganoni et al. (2020) [7] reported that supplementation with sodium phenylbutyrate-taurursodiol resulted in a slower rate of decline in the ALS functional rating scale–Revised (ALSFRS-R) score compared with placebo treatment in ALS patients.

Furthermore, fiber supplementation resulted in higher fecal levels of SCFAs in rats with colitis than in control animals, suggesting that the gut microbial composition and SCFA production depend on the consumption of fiber [8]. Nelson et al. (2000) [9] showed that the risk of ALS onset was negatively associated with the intake of total, soluble, and insoluble fiber. In addition, fiber intake was positively associated with the ALSFRS-R score in ALS patients [10].

Our previous study also reported that Korean ALS patients who consumed low amounts of dietary fiber compared with the dietary reference intake for Koreans had lower ALSFRS-R scores [11]. In addition, we showed that the intake of fruits and beta carotene, a marker for the intake of fruits and vegetables, was negatively associated with the risk of ALS onset [12], and that nutritional status was associated with disease progression and survival time in ALS patients [11,13,14]. However, there has been no study on the relationship between the intake of dietary fiber and ALS prognosis. Therefore, the present study investigated whether dietary fiber intake was related to disease progression rate and survival time. In addition, total fiber intake was divided according to five major fiber-rich foods: vegetables, fruits, grains, legumes, and nuts/seeds. This study also examined whether the intake of dietary fiber was correlated with cytokine levels in the cerebrospinal fluid (CSF) of ALS patients.

## 2. Materials and Methods

### 2.1. Participants

Participants were recruited from the ALS outpatient clinic at Hanyang University Hospital, Seoul, Korea, between March 2011 and September 2019, and were followed until the occurrence of events or the end of September 2020, whichever came first. The events included percutaneous endoscopic gastrostomy (PEG), tracheostomy, and death. Participants who were still alive at five years without the occurrence of events or who were lost to follow-up were censored at the study termination or last visit, respectively.

From the data of currently ongoing series of dietary survey projects on motor neuron diseases, we recruited a total of 480 participants who had undertaken a dietary survey and were diagnosed as definite, clinically probable, or probable laboratory-supported sporadic ALS according to the EI Escorial criteria [15] (Figure 1). At the time of the dietary survey, participants were excluded from the dietary survey projects if they met any of the following criteria: (1) patients whose symptom onset or duration was more than two years (*n* = 150); (2) patients who had previous severe swallowing problems or PEG (*n* = 38); and (3) patients who had undergone tracheostomy or noninvasive ventilation (*n* = 20). Ultimately, a total of 272 participants were included in the analysis.

The study was conducted in accordance with the Declaration of Helsinki, and all procedures were approved by the institutional review board of Hanyang University (HYI-14-105-14). Prior to enrollment, written informed consent was obtained from all the participants or from their caregivers if they were unable to write.

### 2.2. Data Collection

The medical records of participants were reviewed to obtain data on the age at symptom onset, sex, site of symptom onset, height, weight, Korean version of ALSFRS-R scores (K-ALSFRS-R) [16], and the dates of symptom onset, diagnosis, PEG, tracheostomy, and death. A trained dietitian measured a 24-h recall and analyzed the data (CAN-pro version 5.0, Korean Nutrition Society, Seoul, Korea). Additional information on exercise, smoking status, drinking habits, and sun exposure was obtained through interviews. The time point of the dietary survey from symptom onset was defined as the time between symptom onset and the dietary survey. The disease progression rate (∆FS) was calculated using the following formula [17,18]: ∆FS = (48 − ALSFRS-R score at the time of survey/duration from symptom onset to the time of the survey (months)).

### 2.3. Biochemical Assessment

For the evaluation of biochemical components related to nutritional status, a retrospective analysis of the fasting status of blood chemistry was performed. The data from the biochemical study were confined to only the sample data not exceeding the two-month gap from the time of the diet survey. The fasting plasma glucose (FPG), total protein, blood urea nitrogen (BUN), albumin, creatinine, liver enzymes, and blood lipid profiles were measured using an automatic biochemical analyzer (Hitachi 7600 automatic analyzer, Hitachi Ltd., Tokyo, Japan). The complete blood counts were also analyzed using an automated blood cell counter (Sysmex XE-2100, Sysmex Ltd., Kobe, Japan).

An analysis of CSF cytokines was carried out in stored samples of 27 participants whose lumbar puncture for the initial diagnostic purpose was obtainable within six months of the time of the dietary survey. The CSF was collected in polypropylene tubes, centrifuged at 1750× *g* for 10 min at 4 °C, aliquoted into freestanding sterile cryogenic polypropylene conical microtubes, frozen within 30 min of collection, and stored at −80 °C. Cytokine levels, including transforming growth factor (TGF)-β1, interleukin (IL)-6, IL-8, IL-10, IL-1β, tumor necrosis factor (TNF)-α, and monocyte chemoattractant protein-1 (MCP-1), were analyzed using a commercially available cytokine assay kit (Millipore, Billerica, MA), according to the manufacturer’s protocol.

### 2.4. Statistical Analyses

Data analyses were performed using SPSS software version 24.0 (SPSS Inc., Chicago, IL, USA), and a *p*-value of <0.05 was regarded as statistically significant. Variables with a normal distribution were analyzed using the Kolmogorov–Smirnov test, and nonparametric variables were analyzed using nonparametric tests. Continuous variables were presented as means ± standard deviations (SDs) and verified using Bonferroni’s post hoc test after using one-way analysis of variance (ANOVA) or the Kruskal–Wallis test. The proportions of nominal variables were compared using the chi-square test or Fisher’s exact test.

The distribution of the ∆FS in the tertiles of fiber intake was compared using the chi-square test. The Kaplan–Meier method was used to calculate the cumulative survival probabilities, and the difference between the survival curves was assessed using the log-rank test. Correlations between the levels of CSF cytokines and intake of vegetable fiber were calculated using Spearman’s rank correlation coefficients. The hazard ratios and 95% confidence intervals were calculated using the Cox proportional hazards regression analysis after adjusting for confounding factors. The *p*-value for trend was calculated using the median value from the tertiles of dietary fiber intake, and the lowest tertile served as the reference group.

To find the confounding factors for the occurrence of events, Cox regression analysis was performed according to the potential covariates and events. The covariates showing *p*-values < 0.20 in the multivariate models were selected as the confounding factors (age at symptom onset, sex, BMI, drinking habits, disease progression rate, and energy intake) and included in the fully adjusted model [19]. For dietary fiber intake and blood biochemical evaluation, ranked analysis of covariance (ANCOVA) was performed after adjusting for confounding factors.

## 3. Results

### 3.1. Demographics and Clinical Features of Participants According to Total Fiber Intake

Participants in the highest tertile of total fiber intake had a higher rate of exercise and sun exposure, a lower ALSFRS-R score, and a slower ∆FS than those in the middle and lowest tertiles of total fiber intake (Table 1). Participants with a higher intake of total fiber consumed more fiber from vegetables, fruits, legumes, and nuts/seeds (Appendix A).

There were no significant differences in the age at symptom onset, sex, site of initial symptoms, time point of dietary survey from symptom onset, bulbar score, BMI, smoking status, and drinking status among the tertiles of total fiber intake. The levels of FPG, total protein, albumin, creatinine, BUN, alkaline phosphate (ALP), aspartate aminotransferase (AST), alanine aminotransferase (ALT), white blood cell (WBC), hemoglobin, hematocrit, lymphocyte, total cholesterol (TC), triglyceride (TG), HDL-cholesterol (HDL-C), and LDL-cholesterol (LDL-C) were not significantly different among the tertiles of total fiber intake (Appendix A).

### 3.2. Relationship between the Intake of Total Fiber and Prognosis

The mean ∆FS among all the participants was 0.75 at the time point of the survey (Figure 2). The participants with ∆FS values lower than the mean ∆FS value were predominantly (65%) in the highest tertile of vegetable fiber intake, while 44% of these participants were in the lowest tertile (Figure 2). When comparing survival time using Kaplan–Meier survival curves, there was no significant difference in the mean survival time according to the intake of total fiber (Figure 3).

The participants with ∆FS values lower than the mean ∆FS value were also predominant in the highest tertile of nut/seed fiber intake but not in that of fruit, grain, or legume fiber intake (Figure 2). The Kaplan–Meier analysis showed that there were no significant differences in the mean survival time according to the intake of fruit, grain, legume, and nut/seed fiber (Figure 3).

### 3.3. Relationship between the Intake of Vegetable Fiber and Prognosis

Participants with ∆FS values lower than the mean ∆FS value were predominantly (69%) in the highest tertile of vegetable fiber intake; 42% of these participants were in the lowest tertile (Figure 2). In addition, the Kaplan–Meier analysis showed that there was a significant difference in the mean survival time according to the intake of vegetable fiber (Figure 3). In particular, the participants in the highest tertile for vegetable fiber intake had a longer survival time than those in the lowest tertile.

Consistently, the intake of vegetable fiber was negatively correlated with pro-inflammatory cytokines such as IL-1β, IL-6, and MCP-1, suggesting that patients with a higher intake of vegetable fiber had lower levels of pro-inflammatory cytokines (Figure 4). In addition, the intake of vegetable fiber was negatively associated with the risk for events after adjusting for confounding factors in the multivariable-adjusted Cox regression analysis (Appendix A).

## 4. Discussion

This was the first study showing that the intake of vegetable fiber was negatively related to the ∆FS and shorter survival time in patients with ALS. In the present longitudinal cohort study, the mean ∆FS was 0.75, and the mean survival time was 32 months (range: 7–60 months). The mean ∆FS in the present study was similar to that in studies in Japan (0.81) [20], Korea (0.73) [21], and Australia (0.67) [22]. The five-year rate of events, including tracheostomy and death, was reported as 71.3%, and the mean tracheostomy-free survival time was 45.2 months in a Korean study using data from the Korean National Health Insurance Service [23]. However, the rate of events, including PEG, tracheostomy, and death, was 61.0%, and the mean survival time was 32 months during the five-year follow-up in the present study. The present study, using data from a single hospital, showed a lower rate of events than a study using data from the Korean National Health Insurance Service [23], which might be due to the heterogeneity of the database. In addition, the participants in the highest tertile of total fiber intake had a higher rate of exercise and sun exposure in the present study, which might be due to differences in the ALSFRS-R scores. Consistently, our previous study showed that ALS patients with higher ALSFRS-R scores exercised more regularly and had more sun exposure than those in the middle and lowest tertiles of the ALSFRS-R score [11]. Although the ALSFRS-R scores were different among the tertiles of total fiber intake, there were no differences in BMI or the bulbar score, which reflected the ability to eat at the time point of the dietary survey.

In the present study, there was a higher proportion of participants with a ∆FS ≤ 0.75 in the highest tertile of total fiber intake (>19.84 g/d) than in the lowest tertile (≤13.45 g/d). Koreans consumed a high intake of fiber-rich foods such as kimchi, multigrain rice, and fermented soybean paste [24]. The Korea National Health and Nutrition Examination Survey showed that Korean adults consumed 23.2 g/d of dietary fiber [25], which was higher than in America (17.2 g/d) [26], Spain (17.9 g/d) [27], and Japan (15.0 g/d) [28]. However, ALS patients consumed 18.07 g/d of dietary fiber in the present study, which was lower than the general Korean population and the daily recommended intake for Koreans. Since the daily recommended intake for fiber is 20–25 g/d according to the dietary reference intake for Koreans [29], our results suggest that at least 20 g/d of dietary fiber could be recommended for ALS patients. Consistently, our previous study showed that ALS patients consuming lesser amounts of dietary fiber than the dietary reference intake for Koreans had lower ALSFRS-R scores [11]. Dietary patterns, including the intakes of fiber-rich foods such as nuts, seeds, and vegetables, were also positively associated with ALSFRS-R scores in ALS patients [10]. In addition, previous epidemiologic studies have shown that the intake of vegetables, whole grains, nuts, and seeds was negatively associated with the disease progression rate in patients with neurodegenerative diseases such as Parkinson’s disease [30] and multiple sclerosis [31]. It was observed that a treatment with butyrate, the main product of fiber fermentation, delayed disease progression by modulating intestinal homeostasis [6].

Considering the intake of total fiber, the proportion of participants with a ∆FS ≤ 0.75 was greater in the highest tertile of vegetable fiber intake (>10.25 g/d) than in the lowest tertile (≤5.50 g/d). In addition, when comparing survival time using Kaplan–Meier survival curves, there was a significant difference in the survival time according to the intake of vegetable fiber. A longer survival time was observed for participants in the highest tertile of vegetable fiber intake than for those in the lowest tertile. Consistent with the present study, previous epidemiologic studies reported that vegetable fiber intake was negatively associated with mortality in a prospective cohort study [32] and in patients with diseases such as colorectal cancer [33], breast cancer [34], cardiovascular disease, and respiratory disease [35]. In addition, the intake of vegetable fiber was negatively associated with the risk for events including PEG, tracheostomy, and death in the present study. We previously reported that ALS patients consuming more vegetables had higher ALSFRS-R scores, which measure physical performance including bulbar, respiratory, and motor function in ALS patients [11]. Zhang et al. (2019) [36] also showed that a vegetable-rich dietary pattern was negatively associated with the incidence of functional disability, including tube feeding, mental function, and mobility dependence in older adults.

Vegetable fiber, known as fermentable fiber, yields more butyrate than the fiber from cereal bran and fruits in vitro using human fecal microbiota [37,38]. Yang et al. (2014) [39] also reported that the fecal levels of butyrate were associated with the intake of vegetables but not with the intake of grain, fruit, soy products, nuts, and seeds in individuals without digestive diseases, suggesting that butyrate production through the fermentation of fiber varied depending on its source. Vegetable fiber supplementation (e.g., inulin supplementation) shifted the gastrointestinal microbiota composition and was positively associated with fecal butyrate levels [40]. Previous studies have reported that ALS patients had a low Firmicutes/Bacteroidetes ratio, an indicator of gut dysbiosis, and lower levels of butyrate-producing bacteria than healthy controls [4,5]. In addition, ALS mice had impaired intestinal homeostasis and rapid disease progression, which contributed to shorter survival compared with that of wild-type mice [41]. In ALS mice, treatment with sodium phenylbutyrate significantly extended survival and increased body weight and motor performance compared with placebo treatment [42]. A clinical trial also showed that supplementation with sodium phenylbutyrate-taurursodiol resulted in a slower decline of the ALSFRS-R score over a period of 24 weeks in ALS patients [7].

A negative correlation was consistently reported between the intake of vegetable fiber and the levels of pro-inflammatory cytokines, including IL-1β, IL-6, and MCP-1. Hijová et al. (2013) [43] reported that rats with colorectal cancer that were fed a fiber-enriched diet containing inulin from vegetables showed higher fecal levels of butyrate and lower colonic levels of TNF-α than control animals fed a low-fiber diet. Additionally, clinical trials showed that fructo-oligosaccharides, a type of fiber that is abundant in vegetables, reduced the IL-6 levels of intestinal dendritic cells in patients with active Crohn’s disease compared with placebo treatment [44]. Moreover, vegetable intake was negatively associated with serum levels of C-reactive protein, IL-6, and TNF-α in healthy individuals without cardiovascular disease or diabetes mellitus [45].

The present study has a few limitations. First, fiber intake was measured only once at the time of the survey, which might not be reflective of the participants’ long-term dietary intake. Second, only 27 CSF samples from 272 participants were available for the analysis of CSF cytokines; thus, the related findings could not be representative of all ALS patients. Lastly, although adjustments were made for confounders, unmeasured factors could have affected the results of this study.

## 5. Conclusions

The present study demonstrated that the intake of vegetable fiber could influence the disease progression rate and survival time in Korean patients with ALS. In addition, vegetable fiber intake was negatively correlated with pro-inflammatory cytokines, suggesting that the intake of vegetable fiber might delay disease progression and prolong the survival time in Korean ALS patients through an anti-inflammatory effect. Further studies are needed to confirm whether increased intake of fiber during follow-up in the prospective cohort or supplementation of fiber in clinical trial improves the prognosis of ALS.

## Figures and Tables

**Figure 1 nutrients-12-03420-f001:**
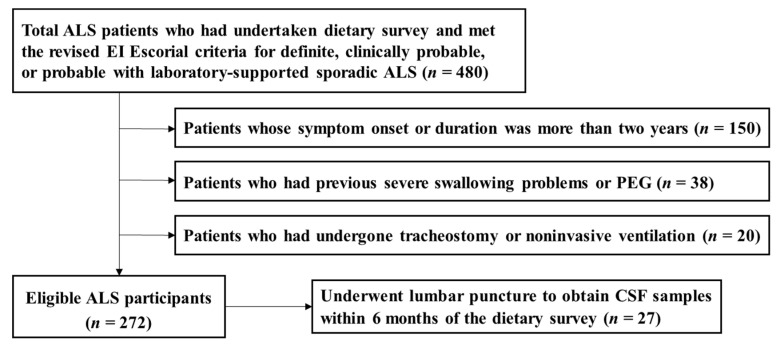
Flow chart for the selection of study participants with amyotrophic lateral sclerosis (ALS). PEG: percutaneous endoscopic gastrostomy; CSF: cerebrospinal fluid.

**Figure 2 nutrients-12-03420-f002:**
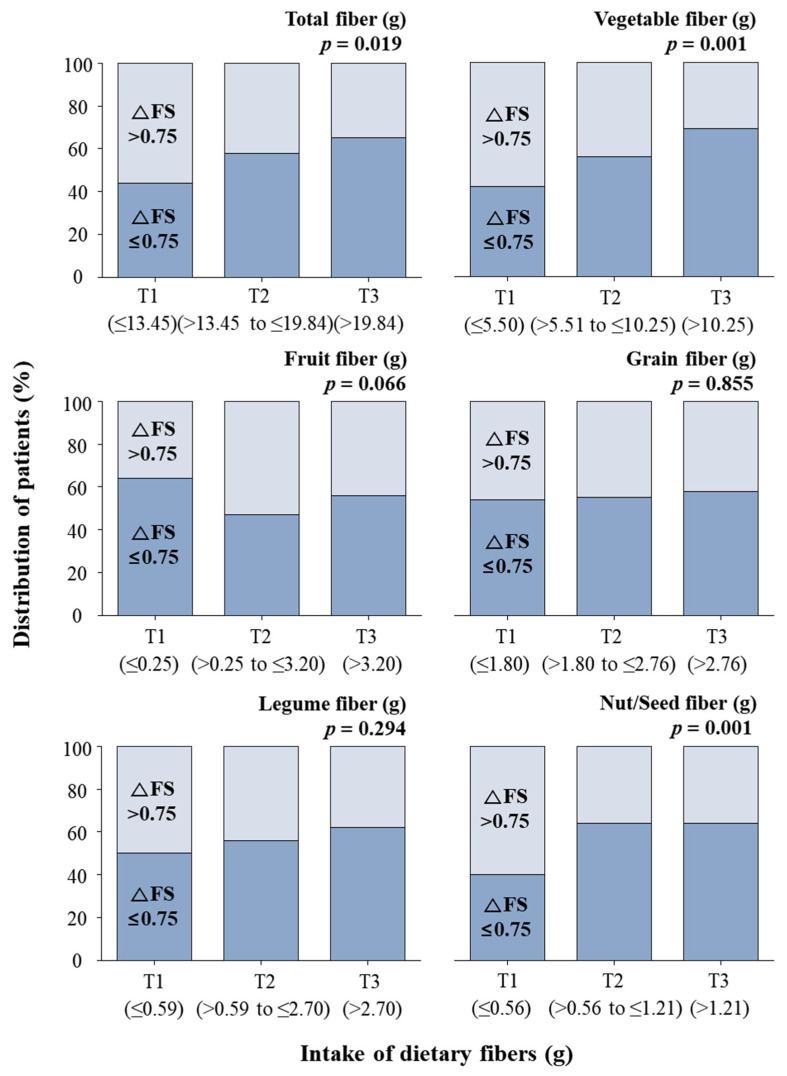
Difference in the distribution of participants with a ∆FS > 0.75 and ∆FS ≤ 0.75 according to tertiles of dietary fiber intake using the chi-square test. ∆FS: disease progression rate.

**Figure 3 nutrients-12-03420-f003:**
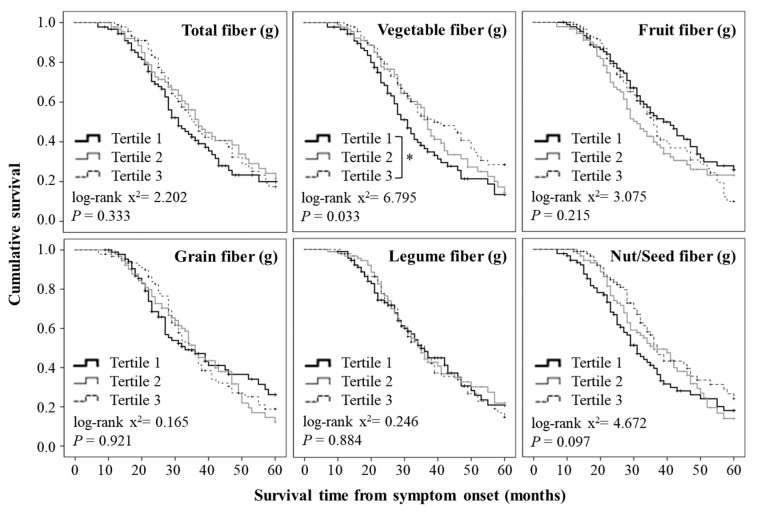
Kaplan–Meier survival curves showing 60-month survival time according to the tertiles of dietary fiber intake. Statistical significance was determined using the log-rank test and Kaplan–Meier method. * *p* < 0.05.

**Figure 4 nutrients-12-03420-f004:**
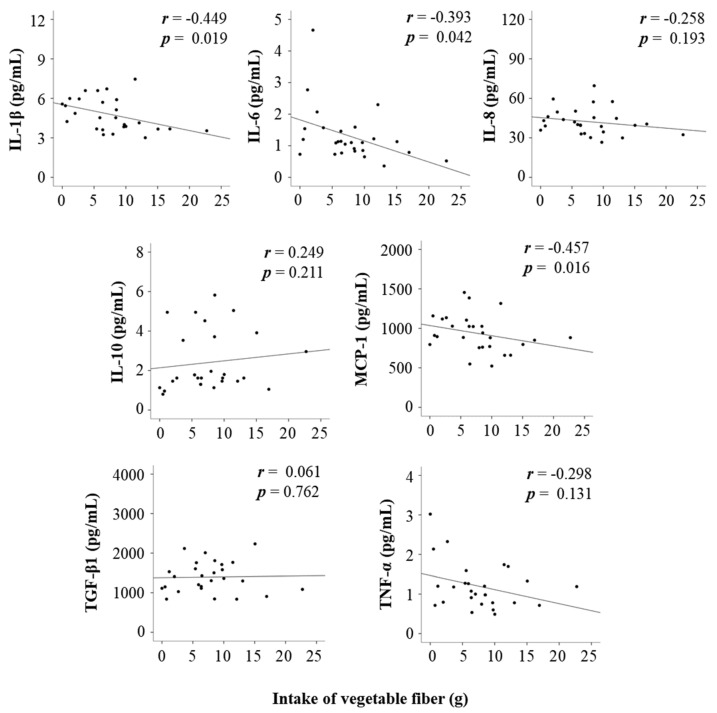
Correlations between the levels of cytokines in cerebrospinal fluid and the intake of vegetable fiber. IL: interleukin; MCP: monocyte chemoattractant protein; TGF: transforming growth factor; TNF: tumor necrosis factor.

**Table 1 nutrients-12-03420-t001:** Demographics and clinical features of the participants according to tertiles of total fiber intake. ^1^

Variables	Total (*n* = 272)	Tertiles of Total Fiber Intake (g)			*p*-Value ^2^
		T1 (*n* = 90) ≤13.45	T2 (*n* = 91) >13.45 to ≤19.84	T3 (*n* = 91) >19.84	
Total fiber intake (g)	18.07 ± 8.75	9.62 ± 2.58 ^3a^	16.51 ± 1.61 ^b^	27.99 ± 6.92 ^c^	<0.001
Age at symptom onset (years)	54.34 ± 10.50	55.29 ± 10.62	54.76 ± 1.66	52.99 ± 10.20	0.404
Male, *n* (%)	145 (53.3)	51 (56.7)	50 (54.9)	44 (48.4)	0.496
Bulbar onset, *n* (%)	59 (21.7)	20 (22.2)	18 (19.8)	21 (23.1)	0.942
Time point of dietary survey from symptom onset (months)	15.31 ± 4.89	15.67 ± 4.74	15.45 ± 5.02	14.81 ± 4.92	0.481
Disease progression rate	0.75 ± 4.89	0.86 ± 0.50 ^3a^	0.74 ± 0.47 ^a,b^	0.65 ± 0.37 ^b^	0.014
ALSFRS-R score ^4^	37.32 ± 6.04	35.34 ± 6.91 ^3a^	37.59 ± 5.67 ^b^	39.01 ± 4.84 ^b^	<0.001
Bulbar score	9.97 ± 1.95	9.61 ± 1.98	10.12 ± 1.74	10.18 ± 2.08	0.050
BMI (kg/m^2^) ^4^	22.73 ± 2.91	22.16 ± 3.03	22.96 ± 2.93	23.05 ± 2.73	0.075
Exercise, *n* (%) ^4^	154 (56.6)	38 (42.2)	53 (58.2)	63 (69.2)	0.001
Sun exposure, *n* (%) ^4^					0.002
Never	72 (26.5)	37 (41.1)	17 (18.7)	18 (19.8)	
<30 min	82 (30.1)	27 (30.0)	28 (30.8)	27 (29.7)	
≥30 min	118 (43.4)	26 (28.9)	46 (50.5)	46 (50.5)	
Smoking, *n* (%) ^4^	28 (10.3)	13 (14.4)	10 (11.0)	5 (5.6)	0.142
Drinking, *n* (%) ^5^	41 (15.1)	14 (15.6)	17 (18.7)	10 (11.0)	0.345

ALSFRS-R: amyotrophic lateral sclerosis functional rating scale-revised; BMI: body mass index. ^1^ Values are presented as means ± SDs or number of participants (percentage distribution), as appropriate; ^2^
*p*-values were calculated using the Kruskal–Wallis test for total fiber intake, age at symptom onset, time point of dietary survey from symptom onset, ALSFRS-R score, and disease progression rate or one-way ANOVA for BMI (followed by Bonferroni’s post hoc test for continuous variables); ^3^ values with different superscript letters in the same row are significantly different at *p* < 0.05 according to ranked ANOVA with Bonferroni’s post hoc test; ^4^ values were investigated at the time point of the dietary survey; exercise = exercising ≥3 times/week; ^5^ drinking = drinking ≥1 time/month.

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
