# Peer review of "Relationship between Dietary Fiber Intake and the Prognosis of Amytrophic Lateral Sclerosis in Korea"

_nutrients, 2020, doi:10.3390/nu12113420_

Round 1
Reviewer 1 Report
The manuscript of Yu et al. presents an original and valuable work about the potential positive influence of dietary fiber intake on disease progression rate and survival time of patients diagnosed with amyotrophic lateral sclerosis.
The manuscript fulfills the specific requirements of the journal. The title is specific and concise. The abstract is written in a single paragraph without headings and subheadings and it summarizes the most important findings of the present study. It properly defines the objectives of the work and its significance. In addition, the abstract is self-contained and citation-free, and it follows the required structure (background, methods, results and conclusions) according to the journal rules. The manuscript provides more than three but less than ten specific keywords; concretely five. The Introduction section provides a background which is well-written and includes the aim of the work. The methodologies carried out in the study are well-described and explained. The study sample is representative as it includes an appropriate number of individuals (272 participants). The study protocol was previously approved by the institutional review board of Hanyang University (HYI-14-105-14) and was conducted in accordance with the guidelines of the Declaration of Helsinki, which is an indispensable requirement in human studies. In case of publication, all the procedures carried out in the study could be reproducible as they include enough details. The manuscript provides novel and interesting results which could be valuable for future research. As an example, it is the first study that suggests the potential inversely association between the intake of vegetable fiber and disease progression rate and shorter survival time in patients with amyotrophic lateral sclerosis. The results are well-discussed taking account of previous studies, and they could be a welcome starting point for future investigations. All figures and tables are appropriately used for the presentation of the results. The Discussion section provides an appropriate interpretation of the most important results. The present work presents some limitations; however, the authors have included a paragraph with view to explaining it. The references are identified with numbers in square brackets in the text and properly present in the references section following the journal requirements.
The specific comments and suggestions for the improvement of the manuscript are included below:
- A concise cover letter should have been sent with the submission of the manuscript in order to explain certain important points (interest and originality of the findings of the study, reasons for why the work fits the scope of the journal and why should it be published, etc.).
- Lines 49, 51, 207, 207, 215, 216, 233: please, avoid the use of personal pronouns (our, we, us, etc.) and use more impersonal terms such as “the authors”.
- Line 40, 46, 235, 240, 255: please, indicate the year of publication of the articles of Paganoni et al., Nelson et al., Zhang et al., Yang et al. and Hijová et al. For instance: “[…] Additionally, Paganoni et al. (2020) reported that supplementation with sodium phenylbutyrate-taurursodiol […]”.
- The conclusions section must be extended as it partially summarizes and supports the data of the present study. Please, provide a more detailed and complete paragraph.
Author Response
Point 1: A concise cover letter should have been sent with the submission of the manuscript in order to explain certain important points (interest and originality of the findings of the study, reasons for why the work fits the scope of the journal and why should it be published, etc.).
Response 1: Thank you for your comment. We revised cover letter explaining the importance of the study findings and reason to fits the scope of the journal.
Point 2: Lines 49, 51, 207, 207, 215, 216, 233: please, avoid the use of personal pronouns (our, we, us, etc.) and use more impersonal terms such as “the authors”.
Response 2: Thank you for your comment. We changed all to “the authors” accordingly.
Point 3: Line 40, 46, 235, 240, 255: please, indicate the year of publication of the articles of Paganoni et al., Nelson et al., Zhang et al., Yang et al. and Hijová et al. For instance: “[…] Additionally, Paganoni et al. (2020) reported that supplementation with sodium phenylbutyrate-taurursodiol […]”.
Response 3: Sorry for the mistakes. We added the year of publication of the articles in introduction and discussion sections.
Point 4: The conclusions section must be extended as it partially summarizes and supports the data of the present study. Please, provide a more detailed and complete paragraph.
Response 4: Thank you for your suggestion. We revised conclusion section as follows, “The present study demonstrated that the intake of vegetable fiber could influence the disease progression rate and survival time in Korean patients with ALS. In addition, vegetable fiber intake was negatively correlated with pro-inflammatory cytokines, suggesting that intake of vegetable fiber might delay disease progression and prolong the survival time in ALS patients through the anti-inflammatory effect.”
Reviewer 2 Report
The authors present the hypothesis that there might be a relationship between dietary fiber intake and the prognosis of amyotrophic lateral sclerosis. The advantage of the study is that the investigations have been made for a long time from 2011 till 2020. Please consider to add the supporting data in the main text – there is a lot of interesting data inside. Please consider also changing the title and the abstract and point that the studies were performer in Korea. The reason is that the different vegetables and diets, in general, differ in the different world regions. The nutrition habits strongly depend on access to regional food – different in Europe, Asia, Africa, America, etc. The short paragraph of the Korean nutrition habits (few sentences) might improve the text (if you don t like to put it in the main text, please add some general comparison in supporting materials.
I agree with the conclusions that these results are the first milestone for further investigations, but please add a few suggestions for further studies in the conclusions and highlight the main results.
Some minor changes are listed below:
Enlarge Figs.1,3,4
- 88 What does it mean „sun exposure”? Why the sun exposure is so important?
Fig. 2 use colors
L 190 please explain what was the result of the negatively correlated with pro-inflammatory 191 cytokines, etc. – please keep in mind that the paper should be easy to read for readers from different field of science (i.e. food science, etc.)
Author Response
Point 1: The advantage of the study is that the investigations have been made for a long time from 2011 till 2020. Please consider to add the supporting data in the main text – there is a lot of interesting data inside.
Response 1: Thank you for your comment. The present study is from the data of currently on-going series of dietary survey projects on Korean ALS cohort. Previously, we published several paper such as Jin et al, Nutritional Neuroscience, 2014, 17, 104-108; Park et al, Nutrition, 2015, 31, 1362-1367; Lee et al, Nutrition, 2019, 33, 181-186; Kim et al, Nutritional Neuroscience, 2020, 23, 8-15. We are also preparing a few papers using the data from our cohort. Thus, it is unable to include more data in this manuscript.
Point 2: Please consider also changing the title and the abstract and point that the studies were performer in Korea. The reason is that the different vegetables and diets, in general, differ in the different world regions. The nutrition habits strongly depend on access to regional food – different in Europe, Asia, Africa, America, etc. The short paragraph of the Korean nutrition habits (few sentences) might improve the text (if you don t like to put it in the main text, please add some general comparison in supporting materials.
Response 2: We agree with your comment. We added “Korea” in title and throughout the manuscript and discussed dietary fiber intake of Korean in line 207-213.
Point 3: I agree with the conclusions that these results are the first milestone for further investigations, but please add a few suggestions for further studies in the conclusions and highlight the main results.
Response 3: Thank you for your suggestion. We revised the suggestion for further studies as whether increased intake of fiber during follow-up in prospective cohort or supplementation of fiber in clinical trial improves the prognosis of ALS are needed.
Point 4: Enlarge Figs.1,3,4
Response 4: Thank you for your comment. We enlarged Figure 1, 3 and 4.
Point 5: 88 What does it mean „sun exposure”? Why the sun exposure is so important?
Response 5: Thank you for your comment. Sun exposure measured by asking how long participant was exposed to sun daily, never, ≤10 minutes, 10~30 minutes, 30 minutes~1 hour, and ≥1 hour, indicated in Table 1. Previous study showed that sun exposure was negatively associated with risk for incidence of neurodegenerative disease such as multiple sclerosis, Parkinson’s disease, and Alzheimer’s disease (Iacopetta et al. 2020). In addition, our previous study showed that ALS patients who had more sun exposure had higher ALSFRS-R scores (Park et al. 2015).
Point 6: Fig. 2 use colors
Response 6: Thank you for your comment. We revised Figure 2.
Point 7: L 190 please explain what was the result of the negatively correlated with pro-inflammatory 191 cytokines, etc. – please keep in mind that the paper should be easy to read for readers from different field of science (i.e. food science, etc.)
Response 7: Thank you for your comment. We revised the sentence as follows, “the intake of vegetable fiber was negatively correlated with pro-inflammatory cytokines, such as IL-1β, IL-6, and MCP-1, suggesting that patients with higher intake of vegetable fiber had lower levels of pro-inflammatory cytokines.”